# Validation of a Dermatology-Focused Multimodal Large Language Model in Classification of Pigmented Skin Lesions

**DOI:** 10.3390/diagnostics15212808

**Published:** 2025-11-06

**Authors:** Joshua Mijares, Neil Jairath, Andrew Zhang, Syril Keena T. Que

**Affiliations:** 1Department of Dermatology, Indiana University School of Medicine, Indianapolis, IN 46202, USA; 2Ronald O. Perelman Department of Dermatology, New York University Grossman School of Medicine, New York, NY 10016, USA

**Keywords:** artificial intelligence, dermatology, melanoma, skin cancer, machine learning, multimodal large language model

## Abstract

**Background:** Artificial intelligence (AI) has shown significant promise in augmenting diagnostic capabilities across medical specialties. Recent advancements in generative AI allow for synthesis and interpretation of complex clinical data including imaging and patient history to assess disease risk. **Objective:** To evaluate the diagnostic performance of a dermatology-trained multimodal large language model (DermFlow, Delaware, USA) in assessing malignancy risk of pigmented skin lesions. **Methods:** This retrospective study utilized data from 59 patients with 68 biopsy-proven pigmented skin lesions seen at Indiana University clinics from February 2023 to May 2025. De-identified patient histories and clinical images were input into DermFlow, and clinical images only were input into Claude Sonnet 4 (Claude) to generate differential diagnoses. Clinician pre-operative diagnoses were extracted from the clinical note. Assessments were compared to histopathologic diagnoses (gold standard). **Results:** Among 68 clinically concerning pigmented lesions, DermFlow achieved 47.1% top diagnosis accuracy and 92.6% any-diagnosis accuracy, with F1 = 0.948, sensitivity 93.9%, and specificity 89.5% (balanced accuracy 91.7%). Claude had 8.8% top diagnosis and 73.5% any-diagnosis accuracy, F1 = 0.816, sensitivity 81.6%, specificity 52.6% (balanced accuracy 67.1%). Clinicians achieved 38.2% top diagnosis and 72.1% any-diagnosis accuracy, F1 = 0.776, sensitivity 67.3%, specificity 84.2% (balanced accuracy 75.8%). DermFlow recommended biopsy in 95.6% of cases vs. 82.4% for Claude, with multiple pairwise differences favoring DermFlow (*p* < 0.05). **Conclusions:** DermFlow demonstrated comparable or superior diagnostic performance to clinicians and superior performance to Claude in evaluating pigmented skin lesions. Although additional data must be gathered to further validate the model in real clinical settings, these initial findings suggest potential utility for dermatology-trained AI models in clinical practice, particularly in settings with limited dermatologist availability.

## 1. Introduction

Skin cancer represents one of the most common malignancies worldwide, with melanoma being the most lethal form of skin cancer [1]. Early detection and accurate diagnosis of pigmented skin lesions are crucial for optimal patient outcomes, yet diagnostic accuracy varies significantly among healthcare providers [2]. The Health Resources and Services Administration predicts an increase in the gap between the supply and demand of full-time dermatologists in the United States over the next 12 years [3]. A recent study has shown that, in a 25-year period starting in 1991, dermatologist visit rates have increased by 68%, and dermatologist visit length has increased by 39% [4]. Currently, dermatologic complaints account for 20% of all physician visits in the United States [5]. The shortage of dermatologists, particularly in underserved areas, has created a need for innovative patient-facing diagnostic tools that can assist in the evaluation of concerning skin lesions. [6].

Artificial intelligence (AI) has emerged as a promising technology to augment clinical decision-making in dermatology. Traditional AI approaches in dermatology have primarily focused on convolutional neural networks (CNNs) trained on large datasets of skin lesion images [7,8,9]. While convolutional neural networks (CNNs) have demonstrated excellent performance for well-defined lesion classification tasks, they may be limited in conditions requiring integration of clinical context, patient history, and morphological patterns, such as inflammatory skin conditions and complex dermatoses where visual appearance alone is insufficient for accurate diagnosis [10,11,12,13].

The emergence of Transformer architecture has represented a significant methodological advancement beyond CNNs. Unlike CNNs that rely on convolutional operations, Transformers utilize attention mechanisms that can capture long-range dependencies and contextual relationships more effectively. Visual Transformers (ViT) have extended this architecture to image recognition tasks, demonstrating superior performance in medical imaging applications including dermatology. The attention-based processing in Visual Transformers enables more nuanced feature extraction and has proven particularly valuable in capturing complex morphological patterns in skin lesions. Importantly, the Vision Transformer architecture forms the foundation for current multimodal models that can simultaneously process both visual and textual data, creating the technical basis for integrating clinical images with patient history and contextual information.

Recent advancements in generative AI and multimodal large language models (LLMs) have introduced new possibilities for clinical applications. Unlike traditional image-only models such as CNNs, multimodal LLMs can process and integrate diverse data types, including clinical images, patient history, and contextual information, to potentially provide more comprehensive diagnostic assessments. Multimodal LLMs such as SkinGPT-4 and PanDerm have demonstrated more comprehensive diagnostic reasoning than unimodal CNNs, which are limited to visual pattern recognition alone [14,15,16]. Broader medical literature also highlights that multimodal generative AI models can generate narrative reports, synthesize patient histories, and provide tailored recommendations, which addresses the need for integration of clinical context in complex diagnostic scenarios [17,18,19,20].

The development of domain-specific AI models trained on specialized medical data represents a significant evolution from general-purpose AI systems. DermFlow, a proprietary dermatology-trained multimodal LLM, was specifically designed to address the unique challenges of dermatologic diagnosis by incorporating extensive dermatology-specific training data that can both cover a wide range of dermatologic conditions and be integrated end-to-end into clinical workflow.

This study aims to evaluate the diagnostic performance of DermFlow in specifically assessing pigmented skin lesions compared to both clinician assessments and a general-purpose multimodal LLM, Claude Sonnet 4 (Claude) provided only with images. By analyzing real-world clinical cases with histopathologic confirmation, we seek to determine the potential utility of specialized AI models in dermatologic practice. This work makes three key contributions to the field: (1) it provides the first direct comparison of a domain-specific dermatology AI system (DermFlow) versus a general-purpose multimodal LLM (Claude) to quantify the value of specialized training; (2) it offers head-to-head comparison with clinician performance using histopathologically confirmed diagnoses in real-world conditions; and (3) it seeks to demonstrate that systematic clinical history integration may dramatically improve AI diagnostic accuracy with important implications for AI system design and clinical workflow integration.

## 2. Materials and Methods

### 2.1. Study Setting

This retrospective study was conducted at Indiana University Health clinics, including Eskenazi Health and IU Health, from February 2023 to May 2025.

### 2.2. Participants

Images were included in the study if they: (1) were obtained from patients seen at Indiana University Health or affiliated clinics between February 2023 and May 2025 with pigmented skin lesions documented in the electronic medical record (EMR); (2) were taken prior to initial biopsy; (3) were associated with a clinician’s pre-biopsy diagnosis; and (4) had histopathological confirmation.

Images were excluded from the study if they: (1) were of lesions that were not visibly pigmented; (2) were taken after the initial biopsy; (3) were obstructed, of low quality, or blurry; or (4) did not include images where the lesion was readily apparent.

A flowchart of the rigorous screening process is displayed in Figure 1.

**Figure 1 diagnostics-15-02808-f001:**
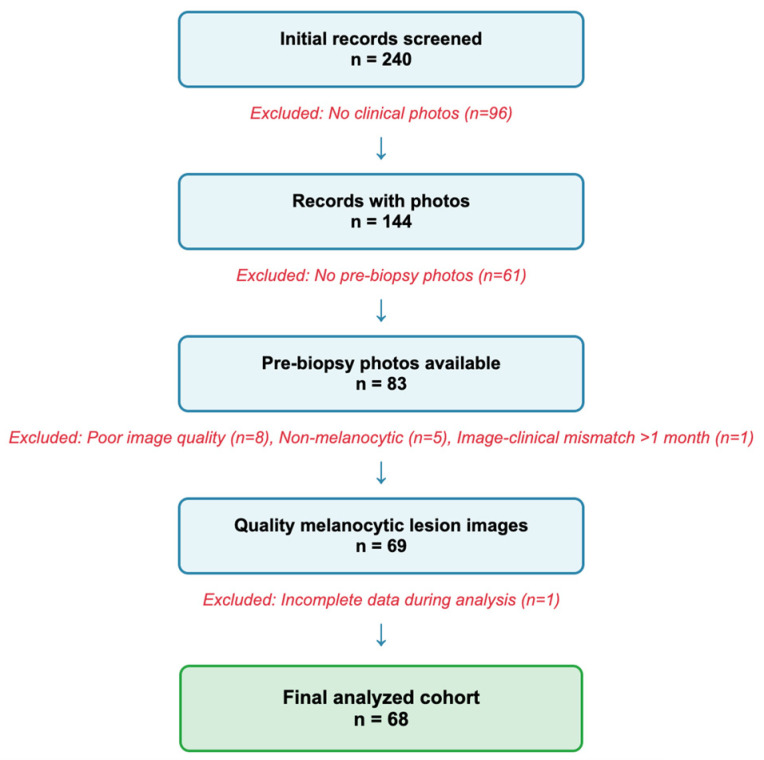
Flowchart of lesion selection process. Numbers indicate individual lesions; 59 unique patients contributed 68 lesions.

Image quality was assessed using standardized criteria including minimum resolution (1280 × 720 pixels), adequate focus with clear lesion borders, appropriate lighting without significant over/underexposure, minimal motion blur, and clear lesion visibility. Quality assessment was performed by DermFlow. No automated pre-processing (e.g., hair removal, vignette correction) was applied to preserve real-world clinical conditions. Only one image per lesion was evaluated by the evaluators. Lesions for which adequate-quality images could not be obtained were excluded from the study. All evaluators (clinician, DermFlow, Claude) assessed identical image sets for each lesion to ensure fair comparison.

### 2.3. Data Collection

Clinical data and clinical images were extracted from the EMR using a standardized data collection form and entered into a secure REDCap database. Variables collected included demographics (age, sex, race/ethnicity), clinical history (personal history of skin cancer, immunosuppressive conditions, family history, etc.), lesion characteristics (location, morphology, duration, changes), clinician differential diagnoses, and histopathologic diagnoses.

### 2.4. AI Model Evaluation

De-identified patient histories and clinical images were input into DermFlow (Delaware, USA), a dermatology-trained multimodal LLM specifically designed for dermatologic applications. Clinical images only were input into Claude, a general-purpose multimodal LLM that serves as the foundation for DermFlow.

Claude was selected as a comparator because (1) it represents state-of-the-art general-purpose multimodal AI, (2) DermFlow is built on Claude’s foundation model, allowing direct assessment of domain-specific training value, and (3) it reflects what clinicians without specialized tools might access. This comparison design isolates the contribution of dermatology-specific training while controlling for underlying model architecture. Both systems utilize transformer-based multimodal architectures capable of processing visual and textual inputs. DermFlow represents a domain-specific adaptation with fine-tuning on dermatological data, while Claude serves as the general-purpose foundation model. To minimize output stochasticity inherent in generative models, Claude was evaluated using its default temperature parameter, while DermFlow employed a temperature setting of 0.2, promoting more deterministic outputs appropriate for clinical applications. Each case was evaluated once. Multiple runs to assess inter-run variability were not performed, representing a study limitation that will be discussed further.

Each model was instructed to output a maximum of 4 differential diagnoses, ranked by likelihood, that were determined to have >85% likelihood. In addition, each model was allowed to provide an additional 1–2 diagnoses that are potentially life-threatening, highly morbid, rapidly progressive, or with potential systemic or other organ involvement (safety diagnoses).

### 2.5. Outcome Measures

The primary outcome was diagnostic accuracy for correctly categorizing lesions as benign, atypical, or malignant, with histopathologic diagnosis serving as the gold standard. Two levels of accuracy were assessed: whether the #1 ranked diagnosis correctly categorized the lesion as benign, atypical, or malignant (top diagnosis accuracy); and whether any diagnosis in the differential correctly categorized the lesion type (any-diagnosis accuracy). This approach focuses on clinically relevant categorization rather than exact specific diagnosis matching, as the critical clinical decision is distinguishing malignant and atypical lesions from benign lesions.

Global accuracy was calculated as the proportion of correct diagnoses among all cases. Balanced accuracy, which accounts for class imbalance by averaging per-class sensitivities, was also calculated to provide comprehensive performance assessment given the imbalanced distribution of diagnosis categories in our clinically enriched cohort.

Secondary outcomes included decision-to-biopsy rates and agreement between AI models and clinicians. Decision-to-biopsy recommendation was determined if the AI model’s differential diagnosis included a diagnosis in the atypical or malignant categories. The clinician’s decision-to-biopsy was 100%, as decision to biopsy was a requirement for inclusion of an image in this study.

### 2.6. Institutional Review Board Statement

The study was conducted in accordance with the Declaration of Helsinki and approved by the Institutional Review Board of Indiana University School of Medicine (IRB #27390, approved 16 May 2025).

### 2.7. Informed Consent Statement

Given the retrospective nature of the study using de-identified data, the requirement for informed consent was waived.

### 2.8. Statistical Analysis

Statistical analyses were performed using IBM SPSS Statistics version 31.0 (IBM Corp., Armonk, NY, USA). Descriptive statistics were calculated for all variables, with categorical variables presented as frequencies and percentages. For diagnostic performance metrics, proportions with 95% confidence intervals were calculated using the Wilson score method, which provides more accurate intervals for proportions near the boundaries (0% or 100%). Statistical significance for comparing diagnostic accuracy proportions between methods was assessed using two-proportion z-tests. Inter-rater agreement was evaluated using Cohen’s kappa coefficient (κ), with interpretation according to Landis and Koch criteria: κ < 0.20 (slight), 0.20–0.40 (fair), 0.40–0.60 (moderate), 0.60–0.80 (substantial), and >0.80 (almost perfect). All statistical tests were two-tailed, and a *p*-value < 0.05 was considered statistically significant.

## 3. Results

### 3.1. Study Population and Clinical Context

This study analyzed a clinically enriched cohort of 68 pigmented lesions that warranted both clinical photography and histopathologic evaluation due to clinical concern for malignancy (Table 1). This represents the population where AI diagnostic assistance would be most clinically valuable—lesions with sufficient clinical suspicion to merit biopsy.

A total of 68 pigmented lesions were analyzed. 49 lesions (72.1%) were histopathologically confirmed as malignant melanoma, 15 lesions (22.1%) classified as atypical, and 4 lesions (5.9%) classified as benign. 39 lesions (57.4%) were indicated with a clinical marker, such as a surgical marking pen, in preparation for biopsy or excision.

### 3.2. Diagnostic Performance

DermFlow demonstrated superior or similar diagnostic accuracy compared to both Claude and clinician assessments across multiple metrics. For top diagnosis accuracy, DermFlow achieved 47.1% (95% CI: 34.8–59.7%) compared to clinicians at 38.2% (95% CI: 26.7–50.8%) and Claude at 8.8% (95% CI: 3.2–17.6%) (Figure 2). When considering any diagnosis within the differential, DermFlow’s performance increased dramatically to 92.6% (95% CI: 84.3–97.1%) outperforming both clinicians at 72.1% (95% CI: 60.8–81.9%) and Claude at 73.5% (95% CI: 61.8–83.4%) (Figure 3). For decision-to-biopsy rates in this high-suspicion cohort, clinicians recommended biopsy for 100% of cases (as this was a requirement for inclusion into the study), DermFlow recommended biopsy for 95.6% (95% CI: 89.4–98.5%) of cases, and Claude recommended biopsy for 82.4% (95% CI: 70.2–90.5%) of cases (Figure 4).

**Figure 2 diagnostics-15-02808-f002:**
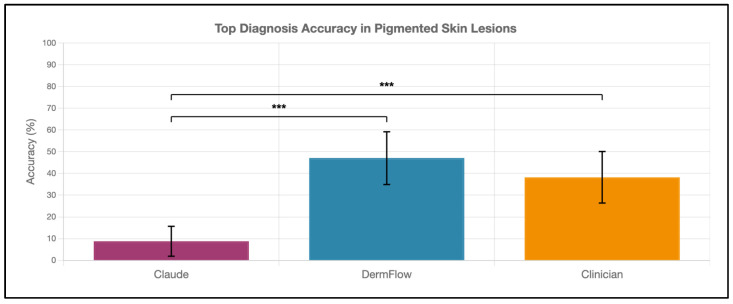
Top diagnosis accuracy of Claude, DermFlow, and clinicians in 68 pigmented skin lesions. Values shown as percentages with 95% confidence intervals. Statistical significance: *** *p* < 0.001.

**Figure 3 diagnostics-15-02808-f003:**
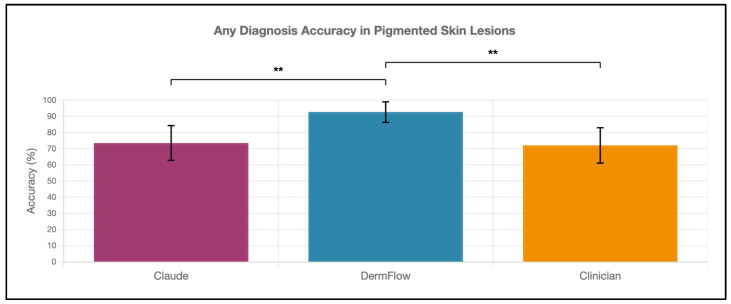
Any-diagnosis accuracy of Claude, DermFlow, and clinicians in pigmented skin lesions. Values shown as percentages with 95% confidence intervals. Statistical significance: ** *p* < 0.01.

**Figure 4 diagnostics-15-02808-f004:**
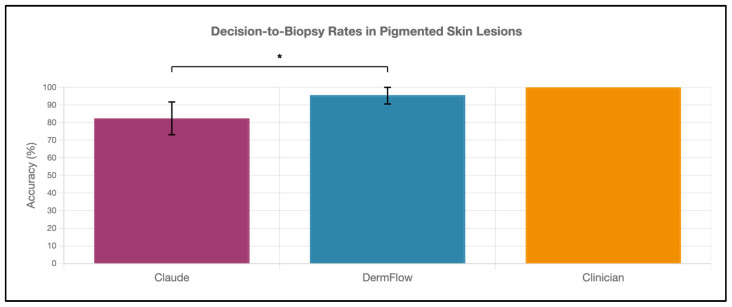
Decision-to-biopsy rates of Claude, DermFlow, and clinicians in pigmented skin lesions. Values shown as percentages with 95% confidence intervals. Statistical significance: * *p* < 0.05.

To provide a complete assessment of diagnostic performance, we calculated comprehensive classification metrics for binary categorization (malignant versus non-malignant lesions), as shown in Table 2. These metrics reveal DermFlow’s excellent diagnostic profile across multiple dimensions.

DermFlow demonstrated excellent performance with both high sensitivity (93.9%, correctly identifying 46 of 49 malignant lesions) and high specificity (89.5%, correctly identifying 17 of 19 non-malignant lesions). This represents a clinically ideal diagnostic profile, avoiding both the danger of missed malignancies and the burden of excessive false positives. The model missed only 3 malignancies (6.1%) and incorrectly flagged only 2 non-malignant lesions (10.5%) as potentially malignant.

DermFlow’s positive predictive value (95.8%) indicates that when it flags a lesion as potentially malignant, it is correct 95.8% of the time, providing high confidence for clinical decision-making. The negative predictive value (85.0%) similarly demonstrates strong reliability when the model categorizes lesions as non-malignant. The F1 score of 0.948 reflects excellent balanced performance across sensitivity and specificity.

In comparison, clinicians showed moderate sensitivity (67.3%, missing 16 of 49 malignancies) with good specificity (84.2%). Claude demonstrated moderate sensitivity (81.6%) but lower specificity (52.6%), incorrectly flagging 9 of 19 non-malignant lesions. DermFlow’s Cohen’s kappa of 0.057, while low in absolute terms, should be interpreted in the context of the challenging diagnostic task and imbalanced dataset.

DermFlow achieved a binary balanced accuracy of 91.7% (average of 93.9% sensitivity and 89.5% specificity), significantly outperforming both clinicians (75.8%) and Claude (67.1%). This metric accounts for the imbalanced class distribution and demonstrates excellent performance on both malignant and non-malignant lesions. For the more granular three-class classification (benign/atypical/malignant based on top diagnosis), DermFlow achieved 41.8% balanced accuracy compared to 57.6% for clinicians and 34.7% for Claude.

The high malignancy prevalence in our dataset (72%) reflects appropriate clinical selection of concerning lesions for biopsy. DermFlow’s ability to maintain both high sensitivity and high specificity in this challenging cohort demonstrates sophisticated diagnostic reasoning rather than conservative over-prediction.

Analysis of misclassification patterns provides important clinical insights. For false negatives (malignant lesions missed): Clinicians missed 16 of 49 malignancies (32.7%), DermFlow missed only 3 of 49 (6.1%), and Claude missed 9 of 49 (18.4%). For false positives (non-malignant lesions incorrectly flagged as potentially malignant): Clinicians incorrectly flagged 3 of 19 non-malignant lesions (15.8%), DermFlow incorrectly flagged 2 of 19 (10.5%), and Claude incorrectly flagged 9 of 19 (47.4%). In dermatologic practice, false negatives carry greater clinical risk as they represent missed cancers with potential for disease progression. DermFlow’s exceptional performance in minimizing false negatives (only 3 missed malignancies) while maintaining low false positives (only 2) represents an optimal balance for clinical decision support. This balanced performance profile distinguishes DermFlow from approaches that achieve high sensitivity only through conservative over-prediction.

### 3.3. Inter-Rater Agreement Analysis

Agreement between AI models and clinicians was assessed using Cohen’s kappa coefficient across multiple diagnostic measures (Table 3). For top diagnosis accuracy, DermFlow showed slight agreement with clinicians (κ = 0.045, 52.9% observed agreement). Agreement between DermFlow and Claude was poor (κ = −0.050, 50.0% observed agreement), indicating performance worse than chance agreement. The highest agreement was observed between clinicians and Claude (κ = 0.196, 67.6% observed agreement), though this remained in the slight agreement range.

Any-diagnosis accuracy agreement patterns could not be assessed because most differential diagnoses produced by clinicians, DermFlow, and Claude included 2 or more diagnostic categories.

For biopsy recommendations, meaningful agreement analysis was only possible between the two AI models since clinicians recommended biopsy for 100% of cases by study design. DermFlow and Claude showed poor agreement for biopsy recommendations (κ = −0.076, 77.9% observed agreement), despite both models having high individual biopsy recommendation rates.

### 3.4. Subgroup Analysis

Performance varied across different patient and lesion characteristics for both top diagnosis accuracy (Table 4) and any-diagnosis accuracy (Table 5). DermFlow maintained superior performance compared to Claude across all subgroups analyzed.

Notable findings include superior performance for head lesions across all methods, with clinicians showing particularly strong performance (top diagnosis accuracy = 71.4%, any-diagnosis accuracy = 92.0%) in this anatomical region. Additionally, the presence of a clinical marker in the analyzed image resulted in higher diagnostic accuracy, although increasing the sample size will be needed to detect significant changes.

#### 3.4.1. Sex-Based Analysis

DermFlow maintained consistent performance across sex, with minimal variation in both top diagnosis accuracy (male: 51.7%, female: 43.6%) and any-diagnosis accuracy (male: 96.6%, female: 89.7%). Clinicians and Claude showed similar consistency across sex groups, suggesting that sex does not significantly influence diagnostic performance for any of the evaluated methods.

#### 3.4.2. Anatomical Location Analysis

Head and face lesions (*n* = 14) demonstrated superior performance across all methods for both accuracy metrics. For top diagnosis accuracy, clinicians achieved their highest performance on head/face lesions (71.4%), followed by DermFlow (64.3%) and Claude (14.3%). This pattern was even more pronounced for any-diagnosis accuracy, where DermFlow achieved perfect performance (100.0%) on head/face lesions, with clinicians (92.9%) and Claude (85.7%) also showing their best regional performance.

Lower extremity lesions (*n* = 19) proved most challenging across all methods, with DermFlow showing its lowest top diagnosis accuracy (21.1%) in this region, though it maintained strong any-diagnosis accuracy (79.0%). This anatomical variation may reflect both imaging challenges in lower extremity photography and the clinical complexity of pigmented lesions in these locations.

#### 3.4.3. Clinical Marker Analysis

The presence of clinical markers in images appeared to benefit AI performance more than clinical assessment. DermFlow showed improved performance when markers were present for both top diagnosis accuracy (53.8% vs. 37.9%) and any-diagnosis accuracy (97.4% vs. 86.2%). Clinician performance remained virtually unchanged regardless of marker presence (38.5% vs. 37.9% for top diagnosis; 76.9% vs. 65.5% for any diagnosis). This finding is expected, as the pre-operative diagnosis is typically made prior to addition of a clinical marker.

#### 3.4.4. Consistency Across Subgroups

DermFlow demonstrated remarkably consistent any-diagnosis accuracy across all analyzed subgroups (range: 87.5% to 100.0%), reinforcing its potential clinical utility across diverse patient populations and lesion characteristics. This consistency is particularly important for clinical implementation, as it suggests reliable performance regardless of patient demographics or lesion location.

## 4. Discussion

### 4.1. Key Findings

Our findings provide compelling evidence for the critical importance of intelligent clinical history integration in AI-assisted dermatologic diagnosis. The dramatic 5-fold difference in top diagnosis accuracy between DermFlow (47.1%) and Claude (8.8%) demonstrates that the ability to systematically gather and integrate clinical context represents a fundamental advancement over image-only AI approaches, regardless of the underlying model sophistication. This interpretation is consistent with previous research that found higher diagnostic accuracy in multimodal AI models compared to unimodal models [18,21,22,23].

While landmark studies by Esteva et al. and Haenssle et al. demonstrated impressive performance with image-only CNN approaches that excel at visual pattern recognition [8,9], our results suggest that diagnostic accuracy may be significantly enhanced when AI systems can systematically integrate clinical context alongside image analysis, addressing a capability gap that purely image-based systems cannot fill, regardless of their visual analysis sophistication.

Perhaps most clinically significant is DermFlow’s exceptional any-diagnosis accuracy of 92.6%, which substantially exceeded both clinicians (72.1%, *p* < 0.01) and Claude (73.5%, *p* < 0.01). This metric reflects the model’s ability to include the correct lesion categorization somewhere within its differential diagnosis, even when not ranked as the top possibility. This comprehensive diagnostic reasoning mirrors how experienced dermatologists approach challenging cases and provides substantial clinical value, as appropriate management can be initiated when the correct diagnosis category is recognized within the differential [14,15,24].

DermFlow’s exceptional any-diagnosis accuracy of 92.6% represents a paradigm shift from previous AI dermatology research, which has been predominantly focused on image classification without clinical context integration [7,8,9,25]. This finding suggests that AI systems capable of intelligent history-taking and clinical reasoning integration can achieve performance levels that approach or exceed the comprehensive diagnostic reasoning that characterizes expert clinical practice.

### 4.2. Clinical Context Integration Is a Fundamental Advance

Our study intentionally compares DermFlow and Claude under their optimal use conditions to answer the clinically relevant question: What performance advantage, if any, does domain-specific AI with clinical history integration provide over general-purpose multimodal AI? The different inputs (DermFlow: images + history; Claude: images only) represent their intended use cases and allow quantification of the combined value of domain specialization and clinical context. This design addresses real-world implementation decisions where healthcare systems must evaluate whether specialized tools justify their cost and complexity versus using readily available general-purpose AI. The 19.1 percentage point difference in any-diagnosis accuracy (92.6% vs. 73.5%) quantifies this added value and provides evidence to inform such decisions.

DermFlow’s achievement of both high sensitivity (93.9%) and high specificity (89.5%) represents a clinically ideal performance profile that is difficult to achieve in diagnostic systems. Many AI diagnostic tools face an inherent sensitivity-specificity tradeoff, where improvements in one metric come at the expense of the other. DermFlow transcends this limitation through sophisticated integration of visual features and clinical context.

The model’s ability to correctly identify 46 of 49 malignancies (missing only 3) while accurately categorizing 17 of 19 non-malignant lesions (incorrectly flagging only 2) demonstrates nuanced diagnostic reasoning. This performance cannot be explained by simple conservative prediction strategies (e.g., flagging everything as potentially malignant), which would yield high sensitivity but poor specificity. Rather, DermFlow appears to synthesize multiple diagnostic features, such as visual morphology, patient history, risk factors, and clinical context, to make refined assessments.

The 95.8% positive predictive value indicates that clinicians can have high confidence in DermFlow’s assessments when it flags potential malignancy. Similarly, the 85.0% negative predictive value suggests reliable performance when categorizing lesions as non-malignant, though appropriate clinical judgment should still guide decisions given the serious consequences of missed melanomas.

This balanced performance profile, combined with the 92.6% overall accuracy and F1 score of 0.948, positions DermFlow as a potentially valuable clinical decision support tool. The model could assist in triage decisions, provide second opinions in challenging cases, and potentially extend expert-level diagnostic reasoning to settings with limited dermatologic expertise.

The superior performance of DermFlow compared to Claude supports our primary working hypothesis that systematic clinical context integration would enhance diagnostic accuracy beyond what is achievable with image analysis alone. The magnitude of improvement suggests several important insights about the nature of dermatologic diagnosis and the role of clinical reasoning in expert performance.

Importantly, our study design allows us to isolate the specific contribution of clinical context integration, as both systems utilize sophisticated multimodal AI capabilities for image analysis. The performance difference therefore reflects the value added by systematic history-taking and clinical reasoning integration rather than differences in underlying AI sophistication or training datasets.

This finding aligns with established principles from cognitive psychology research, demonstrating that expert clinical reasoning relies heavily on pattern recognition combined with systematic integration of contextual information [26,27,28,29,30].The ability of DermFlow to systematically gather and integrate clinical history, including lesion duration, changes over time, family history, and patient risk factors, mirrors the comprehensive assessment approach that characterizes expert dermatologic practice.

### 4.3. Implications for Clinical Practice and Diagnostic Workflows

These results have significant implications for clinical practice that extend beyond dermatology to broader questions of AI integration in clinical diagnosis. The demonstrated value of systematic clinical context integration suggests potential applications that could address current challenges in healthcare delivery.

#### 4.3.1. Systematic History-Taking Enhancement

DermFlow’s intelligent history-taking capabilities could serve as a clinical decision support tool that ensures comprehensive data collection even in busy clinical environments. Unlike human clinicians, who may inadvertently omit important history elements due to time constraints or cognitive load, AI systems can systematically ensure that all relevant clinical context is gathered and appropriately weighted in diagnostic reasoning.

#### 4.3.2. Addressing Geographic and Expertise Disparities

The superior performance of AI-assisted diagnosis with intelligent history-taking suggests potential for extending expert-level diagnostic reasoning to settings with limited dermatologic expertise. Primary care providers could leverage such systems to ensure comprehensive assessment of concerning lesions, potentially reducing diagnostic delays and improving triage decisions.

### 4.4. Limitations

The high prevalence of malignancy in our cohort (72.1% melanoma) reflects selection bias due to pre-selection of concerning lesions and differs significantly from typical dermatology screening populations, where melanoma prevalence is 0.1–0.2% [31]. This enriched dataset provides a rigorous test of diagnostic performance in clinically challenging scenarios where decisions have the highest stakes. Importantly, DermFlow’s ability to achieve both high sensitivity and high specificity in this challenging cohort suggests robust diagnostic reasoning rather than reliance on simple prediction heuristics. In lower-prevalence screening populations, positive predictive value would be expected to decrease while negative predictive value would increase, though sensitivity and specificity should remain stable. Prospective studies in diverse clinical settings, including lower-prevalence screening populations, are needed to fully characterize performance across the spectrum of clinical applications.

Secondly, the images used in this study were extracted from patient charts. These images were not captured with the intention of analysis of diagnostic accuracy, which differentiates this study from other AI model validation studies and prevents meaningful comparison with these studies [32,33,34].

Thirdly, these results come from academic medical centers in Indiana and may not generalize to other practice settings, depending on notetaking and image-capturing practices by clinicians, other clinical protocols, and specific patient populations.

Fourthly, sample size constraints limit power for subgroup analysis. Only 19 non-malignant cases (15 atypical and 4 benign) severely limit specificity analysis and comparison to malignant cases, and small numbers in some anatomical subgroups (neck *n* = 5) prevent meaningful statistical analysis.

Fifthly, LLMs are known to inherently underperform in diagnosing conditions in which visual cues and patterns are the primary diagnostic determinants, which may explain the overall modest performance levels observed across all methods in this challenging melanoma-heavy cohort, especially in comparison to literature values for the sensitivity of CNNs that suggest outperformance of DermFlow significantly [35,36,37]. However, it is important to note that these CNNs often are restricted to high-quality photographs that may require capture by a clinician, which decreases the models’ utility as a patient-facing tool. Additionally, these validation studies are often conducted with curated images that are captured with the intention of being used for diagnostic validation.

To address these limitations, future studies can include (1) conducting prospective studies in dermatology clinics, where all or most pigmented skin lesions of concern can be assessed in real clinical practice; (2) testing diagnostic accuracy of DermFlow using images and clinical data from other medical institutions outside of Indiana; (3) increasing the sample size to increase power for subgroup analysis; and (4) directly comparing diagnostic accuracy of CNNs and DermFlow using the same image and history dataset.

## 5. Conclusions

DermFlow demonstrated exceptional diagnostic performance in evaluating pigmented skin lesions, achieving both high sensitivity (93.9%) and high specificity (89.5%) with 92.6% overall accuracy. This balanced performance represents clinically ideal diagnostic characteristics. DermFlow significantly outperformed both Claude (73.5% accuracy) and clinicians (72.1% accuracy) in this challenging cohort of concerning lesions. The findings demonstrate that domain-specific AI systems integrating clinical history can achieve sophisticated diagnostic reasoning that transcends simple sensitivity-specificity tradeoffs. However, important limitations must be acknowledged: severe demographic homogeneity (98.5% White patients), single-center retrospective design, modest sample size, and lack of prospective real-world validation. These findings require confirmation through large-scale, multi-center, prospective studies with racially and ethnically diverse patient populations across varied practice settings to ensure equitable performance and assess generalizability. While results are promising and suggest potential clinical utility, they represent early-stage validation requiring further rigorous evaluation before clinical implementation.

## Figures and Tables

**Table 1 diagnostics-15-02808-t001:** Study Population Characteristics.

Characteristic	*n* (%)
**Study Population**	
Clinically Concerning Lesions	68 (100)
**Histopathologic Diagnoses**	
Malignant melanoma	49 (72.1)
Atypical nevus/melanocytic proliferation	15 (22.1)
Benign (no atypia mentioned)	4 (5.9)
**Demographics**	
Male	29 (42.6)
Female	39 (57.4)
**Lesion Location**	
Head (scalp, face, ears)	14 (20.6)
Upper Extremity	12 (17.6)
Trunk	16 (23.5)
Lower Extremity	19 (27.9)
Neck	5 (7.4)
Other	2 (2.9)
**Presence of Clinical Marker**	
Present	39 (57.4)
Not Present	29 (42.6)
**Race/Ethnicity**	
White	67 (98.5)
Non-White	1 (1.5)

**Table 2 diagnostics-15-02808-t002:** Binary Classification Performance Metrics (Malignant vs. Non-Malignant) *.

Metric	Claude	DermFlow	Clinician
**Sensitivity**	81.6% (68.6–90.0%)	93.9% (83.5–97.9%)	67.3% (53.4–78.8%)
**Specificity**	52.6% (31.7–72.7%)	89.5% (68.6–97.1%)	84.2% (62.4–94.5%)
**PPV**	81.6% (68.6–90.0%)	95.8% (86.0–98.8%)	91.7% (78.2–97.1%)
**NPV**	52.6% (31.7–72.7%)	85.0% (64.0–94.8%)	50.0% (33.6–66.4%)
**F1 Score**	0.816 (0.732–0.890)	0.948 (0.901–0.980)	0.776 (0.691–0.851)
**Cohen’s κ**	0.295 (0.030–0.534)	0.057 (−0.126–0.270)	−0.169 (−0.352–0.076)
**Accuracy**	73.5% (62.0–82.6%)	92.6% (83.9–96.8%)	72.1% (60.4–81.3%)

** All metrics calculated for any-diagnosis accuracy in distinguishing malignant (n = 49) from non-malignant (n = 19) lesions.*

**Table 3 diagnostics-15-02808-t003:** Inter-Rater Agreement Analysis Values ^1^.

Diagnostic Measure	Comparison	Cohen’s κ	Observed Agreement (%)	Agreement Level
**Top Diagnosis**	DermFlow vs. Clinician	0.046 ± 0.119	52.9	Slight
	DermFlow vs. Claude	−0.051 ± 0.071	50.0	Poor
	Clinician vs. Claude	0.197 ± 0.093	67.6	Slight
**Decision-to-Biopsy**	DermFlow vs. Claude	−0.076 ± 0.037	77.9	Poor
	Clinician vs. Others	N/A ^†^	N/A ^†^	N/A ^†^

*^1^ Kappa interpretation: <0.20 (slight), 0.20–0.40 (fair), 0.40–0.60 (moderate), 0.60–0.80 (substantial), >0.80 (almost perfect). ^†^ Calculation is not applicable because clinician biopsy rate = 100% by study inclusion criteria, precluding meaningful agreement analysis.*

**Table 4 diagnostics-15-02808-t004:** Subgroup Analysis for Top Diagnosis Accuracy *^,1^.

Subgroup	*n*	Claude	DermFlow	Clinician
**Sex**				
Male	29	6.9 (0.0–16.7)	51.7 (32.4–71.1)	34.5 (16.1–52.9)
Female	39	10.3 (0.0–20.2)	43.6 (27.3–59.9)	41.0 (24.9–57.2)
**Lesion Location**				
Head (scalp, face, ears)	14	14.3 (0.0–35.3)	64.3 (35.6–93.0)	71.4 (44.4–98.5)
Upper extremity	12	8.3 (0.0–26.7)	58.3 (25.6–91.1)	25.0 (0.0–53.7)
Trunk	16	12.5 (0.0–30.7)	50.0 (22.5–77.5)	25.0 (0.0–48.8)
Lower extremity	19	5.3 (0.0–16.3)	21.1 (0.0–41.2)	36.8 (13.0–60.7)
Neck	5	0.0 (0.0–0.0)	40.0 (0.0–100.0)	20.0 (0.0–75.5)
**Clinical Marker Presence**				
Present	39	12.8 (0.0–23.8)	53.8 (37.5–70.2)	38.5 (22.5–54.4)
Absent	29	3.4 (0.0–10.5)	37.9 (19.2–56.7)	37.9 (19.2–56.7)

** Values shown as percentage (95% confidence interval). ^1^ Statistical testing not performed due to small subgroup sizes and exploratory nature of analysis.*

**Table 5 diagnostics-15-02808-t005:** Subgroup Analysis for Any-Diagnosis Accuracy *^,1^.

Subgroup	*n*	Claude	DermFlow	Clinician
**Sex**				
Male	29	82.8 (68.1–97.4)	96.6 (89.5–100.0)	65.5 (47.1–83.9)
Female	39	66.7 (51.2–82.2)	89.7 (79.8–99.7)	76.9 (63.1–90.8)
**Lesion Location**				
Head (scalp, face, ears)	14	85.7 (64.5–100.0)	100.0 (100.0–100.0)	92.9 (77.4–100.0)
Upper extremity	12	91.7 (73.3–100.0)	100.0 (100.0–100.0)	50.0 (16.8–83.2)
Trunk	16	87.5 (69.3–100.0)	93.8 (80.4–100.0)	56.3 (29.0–83.6)
Lower extremity	19	42.1 (17.7–66.6)	79.0 (58.76–100.0)	79.0 (58.8–99.1)
Neck	5	60.0 (0.0–100.0)	100.0 (100.0–100.0)	80.0 (24.5–100.0)
**Clinical Marker Presence**				
Present	39	71.8 (57.0–86.6)	97.4 (92.3–100.0)	76.9 (63.1–90.8)
Absent	29	75.9 (59.3–92.4)	86.2 (72.9–99.6)	(47.1–83.9)

** Values shown as percentage (95% confidence interval). ^1^ Statistical testing not performed due to small subgroup sizes and exploratory nature of analysis.*

## Data Availability

The raw data supporting the conclusions of this article will be made available by the authors on request.

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
