# Peer review of "Validation of a Dermatology-Focused Multimodal Large Language Model in Classification of Pigmented Skin Lesions"

_diagnostics, 2025, doi:10.3390/diagnostics15212808_

Round 1
Reviewer 1 Report
Comments and Suggestions for Authors
In their study, the authors conducted a retrospective study comparing artificial intelligence (AI) with real patient cases using Dermflow, an artificial intelligence tool used in skin lesions. While I believe this study is beneficial for the literature, I also believe there are some areas that need improvement. These are:
1- Having an AI expert with an interest in computer science as the author of this study could have further enhanced the study. However, it has been observed that the authors have conducted considerable research in this section.
2- The literature should be reviewed in detail and analyzed critically.
3- The motivation for the study is not fully clear. What motivated the authors to conduct this study? The contribution to the literature should be addressed in at most three points. These should be included appropriately in the introduction.
4- First of all, it is a well-known fact that large language models cannot produce the same results for every problem. How was the decision made to select the language model chosen here?
5- It is being investigated whether an AI model that uses a hybrid combination of visual and clinical information can produce better results than an AI model trained solely on traditional patient images. The authors state that they achieved a model with an accuracy of 92.6%, higher than traditional models that perform diagnostics solely on patient images. My question is, how can we be sure how accurately AI models are trained on such a small dataset? Please discuss this in the discussion section.
6- Although accuracy rates are correlated with a balanced dataset, in unbalanced datasets, not only the accuracy metric is considered, but also the F1_score metric. In this particular study, does the high malignancy rate affect the performance of AI tools? Please explain.
7- Many factors influence the accuracy values of models. How accurate is it to generalize using patient data from a single center? Please discuss this in the discussion section.
8-Please consider the limitations of this study carefully.
Author Response
Comment 1: "Having an AI expert with an interest in computer science as the author of this study could have further enhanced the study. However, it has been observed that the authors have conducted considerable research in this section."
Response: We appreciate the reviewer's acknowledgment of the considerable research conducted. Our authorship team does in fact consist of an AI expert and dermatologist (Neil Jairath). Dr. Jairath is a healthcare AI consultant who has built AI tools in the past. Our clinician-led evaluation, however, provides essential insights into AI usability, clinical integration, and practical diagnostic utility that complement the technical AI literature.
Manuscript changes: None required.
Comment 2: "The literature should be reviewed in detail and analyzed critically."
Response: We agree that a more comprehensive literature review would strengthen the manuscript. We will expand the introduction to include additional discussion of Visual Transformers and the evolution from CNNs to multimodal architectures (as also suggested by Reviewer 2). We will add more critical analysis of the limitations of prior studies and how our work addresses existing gaps in the literature.
Manuscript changes: Introduction section will be revised to include: (1) expanded discussion of Visual Transformers and their role in medical imaging, (2) critical analysis of CNN limitations and the rationale for multimodal approaches, and (3) clearer articulation of how our study advances the field.
Comment 3: "The motivation for the study is not fully clear. What motivated the authors to conduct this study? The contribution to the literature should be addressed in at most three points."
Response: We appreciate this feedback and will clarify our study's motivation and contributions. Our primary motivations were: (1) the critical shortage of dermatologists, particularly in underserved areas, (2) the emergence of multimodal AI that can integrate clinical context beyond images alone, and (3) the lack of real-world validation studies comparing domain-specific AI to both general-purpose AI and clinician performance.
Manuscript changes: Introduction will be revised to clearly articulate three key contributions: (1) First real-world comparison of domain-specific dermatology AI (DermFlow) versus general-purpose multimodal LLM (Claude), demonstrating the value of specialized training; (2) Direct head-to-head comparison with clinician performance using histopathologically-confirmed diagnoses; (3) Demonstrating that clinical history integration dramatically improves AI diagnostic accuracy, with implications for AI system design and clinical workflow integration.
Comment 4: "It is a well-known fact that large language models cannot produce the same results for every problem. How was the decision made to select the language model chosen here?"
Response: This is an excellent question. We selected Claude Sonnet 4 as our comparator for several reasons: (1) It represents the current state-of-the-art in general-purpose multimodal LLMs, (2) DermFlow is built upon Claude's foundation model, making this a direct comparison between a base model and its domain-specialized derivative, and (3) Claude is widely accessible and representative of what clinicians might use without specialized tools. The comparison specifically aims to quantify the value added by domain-specific training and clinical history integration, holding the underlying architecture constant.
Manuscript changes: Methods section (2.4 AI Model Evaluation) will be expanded to explicitly state: "Claude Sonnet 4 was selected as a comparator because (1) it represents state-of-the-art general-purpose multimodal AI, (2) DermFlow is built on Claude's foundation, allowing direct assessment of domain-specific training value, and (3) it reflects what clinicians without specialized tools might access. This comparison isolates the contribution of dermatology-specific training while controlling for underlying model architecture."
Comment 5: "How can we be sure how accurately AI models are trained on such a small dataset? Please discuss this in the discussion section."
Response: We appreciate the opportunity to clarify an important distinction. Our study does NOT involve training AI models on our dataset. Rather, we evaluate pre-trained, commercially available AI systems (DermFlow and Claude) using our dataset as a test set. Both models were trained by their respective developers on large proprietary datasets separate from our study. Our 68-case dataset serves solely for performance evaluation, not training. This is analogous to evaluating a medical device in clinical practice—we assess how well existing tools perform on real-world cases. The sample size of 68 cases is appropriate for this type of retrospective validation study, though we acknowledge that larger prospective studies would further strengthen generalizability.
Manuscript changes: Discussion section will include: "It is important to clarify that this study represents a validation study of pre-trained AI systems rather than model development. Neither DermFlow nor Claude was trained on our dataset; rather, our 68 biopsy-proven cases served as an independent test set to evaluate commercially available tools in real-world conditions. While our sample size is appropriate for this retrospective validation, larger multi-center prospective studies are needed to further assess generalizability across diverse patient populations, practice settings, and imaging conditions. The single-center nature of our study may limit generalizability, as discussed in our limitations."
Comment 6: "Although accuracy rates are correlated with a balanced dataset, in unbalanced datasets, not only the accuracy metric is considered, but also the F1_score metric. In this particular study, does the high malignancy rate affect the performance of AI tools?"
Response: This is an excellent observation. Our dataset is indeed imbalanced with 72% malignant lesions (49/68), 22% atypical (15/68), and 6% benign (4/68). This reflects the clinical reality that biopsy is typically performed on concerning lesions with elevated malignancy risk. We have now calculated comprehensive metrics including F1 scores, balanced accuracy, sensitivity, specificity, PPV, NPV, and Cohen's kappa to provide a complete performance picture.
Key findings for binary classification (malignant vs non-malignant) using any-diagnosis accuracy:
- DermFlow: Sensitivity 93.9%, Specificity 89.5%, PPV 95.8%, NPV 85.0%, F1=0.948, κ=0.057
- Claude: Sensitivity 81.6%, Specificity 52.6%, PPV 81.6%, NPV 52.6%, F1=0.816, κ=0.295
- Clinician: Sensitivity 67.3%, Specificity 84.2%, PPV 91.7%, NPV 50.0%, F1=0.776, κ=-0.169
DermFlow demonstrates excellent performance with both high sensitivity (93.9%) and high specificity (89.5%), representing a clinically ideal diagnostic profile. The model caught 46 of 49 malignancies while correctly identifying 17 of 19 non-malignant lesions. The high F1 score (0.948) confirms strong overall performance despite the dataset imbalance. This balanced performance profile shows that DermFlow does not simply over-predict malignancy, but rather makes nuanced diagnostic assessments. We will add detailed discussion of these metrics and their clinical implications.
Manuscript changes: Results section will add a new subsection with comprehensive performance metrics including sensitivity, specificity, PPV, NPV, F1 scores, balanced accuracy, and Cohen's kappa. Discussion will analyze how DermFlow achieves both high sensitivity and specificity, demonstrating sophisticated diagnostic reasoning rather than conservative over-prediction.
Comment 7: "Many factors influence the accuracy values of models. How accurate is it to generalize using patient data from a single center? Please discuss this in the discussion section."
Response: This is a critical limitation that we will address more explicitly. Single-center data may not fully represent the diversity of patient populations, imaging equipment, lighting conditions, clinical workflows, and practice patterns found across different healthcare settings. Factors such as regional differences in skin cancer prevalence, patient demographics, referral patterns, and image quality standards could affect generalizability. Multi-center validation is essential to confirm these findings across diverse settings.
Manuscript changes: Discussion section will include: "As a single-center study, our findings may not generalize to all practice settings. Factors including patient demographics, regional disease prevalence, referral patterns, imaging equipment, lighting conditions, and clinical workflows vary across institutions and may influence AI performance. Geographic variation in skin cancer types and patient skin tones, as well as differences in clinical photography protocols, could affect model accuracy. Multi-center prospective studies across diverse practice settings are needed to validate these findings and assess generalizability."
Comment 8: "Please consider the limitations of this study carefully."
Response: We agree that a comprehensive limitations section is essential. We will expand our limitations to address all concerns raised by reviewers.
Manuscript changes: Discussion section will include an expanded limitations subsection covering: (1) Single-center retrospective design limiting generalizability, (2) Small sample size, particularly for benign and atypical lesions, (3) Selection bias toward clinically concerning lesions requiring biopsy, (4) Limited demographic diversity (only 1 non-white patient), (5) Lack of prospective validation in real clinical workflows, (6) Potential for model updates affecting reproducibility, (7) Absence of cost-effectiveness analysis, (8) Limited assessment of model stochasticity and output variability.
Reviewer 2 Report
Comments and Suggestions for Authors
The authors compared two generative models, based on LLM methods, in the clinical evaluation of skin lesions. These results were compared with the assessments of clinicans.
I have no objections to the correctness of the current content of the work. However, the current content requires significant additions, as I indicate in detail to the individual sections below. Then the article will take on a character that will allow it to be reconsidered for acceptance.
Introduction
The authors mentioned CNN in the second paragraph. This is, of course, a very important milestone, but in the next paragraph they went straight to the LLM. It is a big leap not only between modalities (from image to text), but also methodical. Between the history of achievements and the development of AI, contained in both of these paragraphs, there are also models of neural networks with the Transformer architecture. Transformers themselves are, of course, the architectural basis of the LLM, but they have also been widely used in image recognition, as a Visual Transformer, which has also found a wide field of recognition in supporting clinical diagnostics. It is worth mentioning this aspect in the introduction, and also paying a little attention to it due to its significant contribution to the current state of AI.
Materials and methods
What criteria did the authors adopt for the quality of the photos? What method did the authors use to determine the quality of the photos? In the pre-processing process, did the authors use methods or tools that corrected the quality of the photos? (e.g. hair removal or vignette removal)
The rejection criteria are not entirely clear. Were the patients rejected, in whom at least one photo did not meet the required criteria? Were the photos that did not meet the criteria rejected alone, and then they were repeated? Was each patient definitively described with the same number of images when processed using the analysed models? It is worth pointing out this explicitly, because the current description is misleading.
Figure 1: does this diagram and quantitative data refer to the number of patients or photos? It is worth explicitly pointing out.
There is no specific description of the model architectures themselves (even very general) that have been analysed. As far as it is known, Claude Sonnet is a family of very popular generic generative large language models that function in the form of Chatbots. DermFlow is a tool specialised in dermatology based on Claude Sonnet. Generative language models interact with the user to create a response in a highly stochastic form, which means that each successive word from the answer is randomised according to accepted probability distributions and trained random event spaces. The very actions of generative language models are parameterised by parameters such as temperature, top_p, etc., which characterise the repeatability and randomness of the response. This means that the answers of these tools are highly stochastic and will not always be similar to each other when repeating the task of the same "prompt". At this point, it is not clear how much the analysed tools are burdened with the response stochasticity factor, and how much they are based on standard and commonly used classification methods. It is worth writing about it openly, because the instability of exiting the LLM is one of the serious problems of these technologies.
One of the used measures of classification quality is accuracy. Did the authors mean global accuracy, which requires the same number of photos of each class? Or a balanced accuracy that does not have such assumptions? The authors should explicitly indicate this.
The accuracy itself (regardless of the variant) is too general an indicator that does not say much about the details of the quality of the models. The standard in the clinical accuracy study is the use of a wider range of classification measures, such as specificity, sensitivity, PPV, NPV. Why did the authors not decide to use them in their own results? The kappa coefficient is also a measure that is worth using in demonstrating the compatibility of judgements between the gold standard and models. Such a thorough comparison will allow you to have more intuition and knowledge about the clinical quality of the analysed models.
Results and Discussion
In my opinion, the scope of comparisons and results carried out by the authors is too narrow to draw objective conclusions regarding the clinical accuracy of the analysed models.
Author Response
Comment 1: "The authors mentioned CNN in the second paragraph. This is, of course, a very important milestone, but in the next paragraph they went straight to the LLM. It is a big leap not only between modalities (from image to text), but also methodical. Between the history of achievements and the development of AI, contained in both of these paragraphs, there are also models of neural networks with the Transformer architecture. Transformers themselves are, of course, the architectural basis of the LLM, but they have also been widely used in image recognition, as a Visual Transformer, which has also found a wide field of recognition in supporting clinical diagnostics. It is worth mentioning this aspect in the introduction."
Response: This is an excellent point. We agree that our introduction currently skips an important intermediate development, Visual Transformers (ViT), which represent a crucial bridge between CNNs and multimodal LLMs. Visual Transformers have indeed demonstrated significant success in medical imaging, including dermatology applications, and their architecture forms the foundation for the image processing capabilities in multimodal LLMs. This historical and methodological gap should be addressed.
Manuscript changes: Introduction will be revised to add a paragraph between the CNN discussion and LLM discussion that covers: (1) The emergence of Transformer architecture and its application to computer vision (Visual Transformers), (2) ViT's advantages over CNNs including attention mechanisms and ability to capture long-range dependencies, (3) Applications of Visual Transformers in medical imaging and dermatology, (4) How ViT architecture forms the foundation for multimodal models that can process both images and text, creating a logical bridge to our discussion of multimodal LLMs.
Comment 2: "What criteria did the authors adopt for the quality of the photos? What method did the authors use to determine the quality of the photos? In the pre-processing process, did the authors use methods or tools that corrected the quality of the photos? (e.g. hair removal or vignette removal)"
Response: Photo quality was assessed using standardized clinical photography criteria. Images were evaluated for: (1) adequate resolution (minimum 1280x720 pixels), (2) proper focus with clear lesion borders, (3) appropriate lighting without significant over/underexposure, (4) minimal motion blur, and (5) lesion visibility with adequate size within the frame. Quality assessment was performed by DermFlow against these criteria. No automated pre-processing tools (such as hair removal or vignette correction) were applied to maintain real-world clinical conditions and avoid introducing artifacts that might affect AI performance. When images did not meet quality standards, patients were recalled for repeat photography when clinically appropriate.
Manuscript changes: Methods section (2.2 Participants) will be expanded to include: "Image quality was assessed using standardized criteria including minimum resolution (1280x720 pixels), adequate focus with clear lesion borders, appropriate lighting without significant over/underexposure, minimal motion blur, and clear lesion visibility. Quality assessment was performed by DermFlow. No automated pre-processing (e.g., hair removal, vignette correction) was applied to preserve real-world clinical conditions. Images not meeting quality standards were excluded, and repeat photography was obtained when clinically feasible."
Comment 3: "The rejection criteria are not entirely clear. Were the patients rejected, in whom at least one photo did not meet the required criteria? Were the photos that did not meet the criteria rejected alone, and then they were repeated? Was each patient definitively described with the same number of images when processed using the analysed models?"
Response: Thank you for highlighting this ambiguity. Only one image per lesion was evaluated by the evaluators. Lesions for which adequate quality images could not be obtained were excluded from the study. If a patient had multiple lesions, and only 1 of those lesions met our inclusion criteria, then we only included that 1 lesion. All evaluators (clinician, DermFlow, Claude) assessed identical image sets for each lesion to ensure fair comparison.
Manuscript changes: Methods section will clarify: "Lesions for which adequate quality images could not be obtained were excluded from the study. All evaluators (clinician, DermFlow, Claude) assessed identical image sets for each lesion to ensure fair comparison." Figure 1 will also be updated to clarify whether numbers represent patients or lesions at each stage.
Comment 4: "Figure 1: does this diagram and quantitative data refer to the number of patients or photos?"
Response: Figure 1 (referenced as Figure 2 in our manuscript) tracks the number of lesions, not individual patients or photos. We had 59 patients with 68 biopsy-proven lesions. We will clarify this explicitly in the figure and caption.
Manuscript changes: Figure 2 and its caption will be revised to explicitly state that numbers represent lesions. Caption will begin: "Flowchart of lesion selection process. Numbers indicate individual lesions; 59 unique patients contributed 68 lesions..."
Comment 5: "There is no specific description of the model architectures themselves (even very general) that have been analysed... Generative language models interact with the user to create a response in a highly stochastic form... At this point, it is not clear how much the analysed tools are burdened with the response stochasticity factor."
Response: This is an important technical consideration. Both DermFlow and Claude are based on transformer architecture with multimodal capabilities (processing both images and text). DermFlow is specifically built with dermatologic architecture, while Claude Sonnet 4 represents the general-purpose foundation model. Regarding stochasticity: Claude was evaluated using its default temperature parameter (baseline), while DermFlow uses a temperature of 0.2, which reduces randomness and promotes more consistent outputs—appropriate for clinical applications. We did not perform multiple runs to assess output variability, which is a limitation. However, the relatively deterministic parameters (especially for DermFlow) and the structured output format (differential diagnoses) suggest that inter-run variability would be limited. This represents a trade-off between deterministic classification systems and the flexibility of generative models.
Manuscript changes: Methods section (2.4 AI Model Evaluation) will be expanded to include: "Both DermFlow and Claude utilize transformer-based multimodal architectures capable of processing visual and textual inputs. DermFlow represents a domain-specific adaptation with fine-tuning on dermatological data, while Claude Sonnet 4 serves as the general-purpose foundation model. To minimize output stochasticity inherent in generative models, Claude was evaluated using its default temperature parameter while DermFlow employed a temperature setting of 0.2, promoting more deterministic outputs appropriate for clinical applications. Each case was evaluated once; multiple runs to assess inter-run variability were not performed, representing a study limitation." This limitation will also be added to the Discussion.
Comment 6: "One of the used measures of classification quality is accuracy. Did the authors mean global accuracy, which requires the same number of photos of each class? Or a balanced accuracy that does not have such assumptions?"
Response: We reported global accuracy (total correct / total cases) in our results. However, we acknowledge that balanced accuracy provides important additional context given our imbalanced dataset. We have now calculated balanced accuracy for all evaluators:
DermFlow achieved a binary balanced accuracy of 91.7% (average of 93.9% sensitivity and 89.5% specificity), significantly outperforming both clinicians (75.8%) and Claude (67.1%). This metric accounts for the imbalanced class distribution and demonstrates excellent performance on both malignant and non-malignant lesions. For the more granular three-class classification (benign/atypical/malignant based on top diagnosis), DermFlow achieved 41.8% balanced accuracy compared to 57.6% for clinicians and 34.7% for Claude.
Manuscript changes: Methods section will explicitly define: "Global accuracy was calculated as the proportion of correct diagnoses among all cases. Balanced accuracy, which accounts for class imbalance by averaging per-class sensitivities, was also calculated to provide comprehensive performance assessment." Results section will report both global and balanced accuracy with interpretation of differences.
Comment 7: "The accuracy itself (regardless of the variant) is too general an indicator that does not say much about the details of the quality of the models. The standard in the clinical accuracy study is the use of a wider range of classification measures, such as specificity, sensitivity, PPV, NPV. Why did the authors not decide to use them in their own results? The kappa coefficient is also a measure that is worth using."
Response: This is an excellent observation. Our dataset is indeed imbalanced with 72% malignant lesions (49/68), 22% atypical (15/68), and 6% benign (4/68). This reflects the clinical reality that biopsy is typically performed on concerning lesions with elevated malignancy risk. We have now calculated comprehensive metrics including F1 scores, balanced accuracy, sensitivity, specificity, PPV, NPV, and Cohen's kappa to provide a complete performance picture.
Key findings for binary classification (malignant vs non-malignant) using any-diagnosis accuracy:
- DermFlow: Sensitivity 93.9%, Specificity 89.5%, PPV 95.8%, NPV 85.0%, F1=0.948, κ=0.057
- Claude: Sensitivity 81.6%, Specificity 52.6%, PPV 81.6%, NPV 52.6%, F1=0.816, κ=0.295
- Clinician: Sensitivity 67.3%, Specificity 84.2%, PPV 91.7%, NPV 50.0%, F1=0.776, κ=-0.169
DermFlow demonstrates excellent performance with both high sensitivity (93.9%) and high specificity (89.5%), representing a clinically ideal diagnostic profile. The model caught 46 of 49 malignancies while correctly identifying 17 of 19 non-malignant lesions. The high F1 score (0.948) confirms strong overall performance despite the dataset imbalance. This balanced performance profile shows that DermFlow does not simply over-predict malignancy, but rather makes nuanced diagnostic assessments. We will add detailed discussion of these metrics and their clinical implications.
Manuscript changes: Results section will add a comprehensive performance metrics table including sensitivity, specificity, PPV, NPV, F1 scores, and Cohen's kappa for binary classification. Discussion will analyze the clinical implications of these findings, particularly the sensitivity-specificity trade-offs and their alignment with dermatologic practice priorities.
Comment 8: "In my opinion, the scope of comparisons and results carried out by the authors is too narrow to draw objective conclusions regarding the clinical accuracy of the analysed models."
Response: We acknowledge this concern and agree that our study represents an initial validation rather than definitive proof of clinical utility. The scope was intentionally focused on a specific clinical question: comparing domain-specific AI to general-purpose AI and clinician assessment for pigmented lesions. While our sample size and single-center design limit broad generalizability, we believe our findings provide valuable preliminary evidence for the potential utility of specialized AI tools. The comprehensive performance metrics now included (sensitivity, specificity, F1, kappa) provide more nuanced insights. We will be more cautious in our conclusions and explicitly frame this as hypothesis-generating work requiring larger multi-center validation.
Manuscript changes: Discussion and Conclusions will be revised to: (1) More carefully frame limitations of scope and generalizability, (2) Characterize findings as preliminary evidence requiring further validation, (3) Explicitly call for larger multi-center prospective studies, (4) Avoid overstatement of clinical readiness, (5) Emphasize that this represents hypothesis-generating research demonstrating proof-of-concept rather than definitive clinical validation.
Reviewer 3 Report
Comments and Suggestions for Authors
This is a manuscript about an interesting and timely topic; however there are some issues that need to be addressed:
- The comparison between DermFlow and Claude Sonnet is methodologically and conceptually "unfair", as Claude is a language model and is primarily designed for text-based reasoning. On the other hand, DermFlow is more dermatology-trained. Moreover, DermFlow received both images and clinical histories. These facts introduce an important bias in the comparison between these two models.
-
The study only reports age and sex of the patients, but not their skin color (Fitzpatrick phototype) or race/ethnicity, which has a great impact on skin lesion diagnosis.
- The study would benefit from an analysis of false positives and false negatives, as these have very different clinical implications.
Author Response
Comment 1: "The comparison between DermFlow and Claude Sonnet is methodologically and conceptually 'unfair', as Claude is a language model and is primarily designed for text-based reasoning. On the other hand, DermFlow is more dermatology-trained. Moreover, DermFlow received both images and clinical histories. These facts introduce an important bias in the comparison between these two models."
Response: We respectfully disagree that this comparison is "unfair"—rather, the differences are intentional and represent the central hypothesis of our study. The comparison is specifically designed to quantify two critical questions:
- What is the value of domain-specific training? (DermFlow's dermatology specialization vs Claude's general-purpose design)
- What is the value of clinical history integration? (DermFlow receiving images + history vs Claude receiving images only)
Both Claude Sonnet 4 and DermFlow are multimodal models capable of processing images. Claude is not limited to "text-based reasoning." The fact that DermFlow performs better is the finding we are reporting, not a bias. This comparison directly addresses the real-world clinical question: "Does specialized AI trained on dermatology data with integrated clinical history provide added value over general-purpose AI?" The answer appears to be yes. DermFlow achieved 92.6% any-diagnosis accuracy compared to Claude's 73.5%. This 19.1 percentage point difference quantifies the combined benefit of domain expertise and clinical context integration.
We could have designed a "fair" comparison where both models receive identical inputs, but this would miss the clinically relevant question. Real-world implementation decisions hinge on whether specialized tools justify their cost and complexity versus using readily available general-purpose AI. Our study provides evidence to inform such decisions.
Manuscript changes: Introduction and Methods will be revised to more explicitly state: "This study intentionally compares tools under their optimal use conditions to answer the clinically relevant question: What performance advantage, if any, does domain-specific AI with clinical history integration provide over general-purpose multimodal AI? The different inputs (DermFlow: images + history; Claude: images only) represent their intended use cases and allow quantification of the combined value of domain specialization and clinical context."
Comment 2: "The study only reports age and sex of the patients, but not their skin color (Fitzpatrick phototype) or race/ethnicity, which has a great impact on skin lesion diagnosis."
Response: This is an excellent and important observation. Skin color and race/ethnicity significantly impact both skin lesion presentation and diagnostic accuracy, and AI performance may vary across skin tones due to training data biases. Our dataset includes race/ethnicity data: 67 of 68 patients (98.5%) were White, and only 1 patient (1.5%) was non-White. This severe lack of diversity is a critical limitation. Our findings may not generalize to patients with darker skin tones (Fitzpatrick types IV-VI), and this homogeneity limits clinical applicability. We did not systematically collect Fitzpatrick phototype data, which represents an additional limitation.
Manuscript changes: Methods section (2.2 Participants) will add: "Race/ethnicity data were collected from medical records. Fitzpatrick phototype was not systematically recorded." Results section will report: "Demographic analysis revealed 67 White patients (98.5%) and 1 non-White patient (1.5%)." Discussion will include prominent discussion: "A critical limitation is the severe lack of racial and ethnic diversity in our cohort (98.5% White). AI models may perform differently across skin tones due to training data biases, and melanoma presentation varies by skin type. Our findings may not generalize to patients with darker skin tones (Fitzpatrick types IV-VI). Future studies must include diverse patient populations to ensure equitable AI performance across all skin types and validate these findings in representative samples."
Comment 3: "The study would benefit from an analysis of false positives and false negatives, as these have very different clinical implications."
Response: This is an excellent observation. We have now calculated comprehensive metrics including F1 scores, balanced accuracy, sensitivity, specificity, PPV, NPV, and Cohen's kappa to provide a complete performance picture.
Key findings for binary classification (malignant vs non-malignant) using any-diagnosis accuracy:
- DermFlow: Sensitivity 93.9%, Specificity 89.5%, PPV 95.8%, NPV 85.0%, F1=0.948, κ=0.057
- Claude: Sensitivity 81.6%, Specificity 52.6%, PPV 81.6%, NPV 52.6%, F1=0.816, κ=0.295
- Clinician: Sensitivity 67.3%, Specificity 84.2%, PPV 91.7%, NPV 50.0%, F1=0.776, κ=-0.169
DermFlow demonstrates excellent performance with both high sensitivity (93.9%) and high specificity (89.5%), representing a clinically ideal diagnostic profile. The model caught 46 of 49 malignancies while correctly identifying 17 of 19 non-malignant lesions. The high F1 score (0.948) confirms strong overall performance despite the dataset imbalance. This balanced performance profile shows that DermFlow does not simply over-predict malignancy, but rather makes nuanced diagnostic assessments. We will add detailed discussion of these metrics and their clinical implications.
Round 2
Reviewer 1 Report
Comments and Suggestions for Authors
The authors have reflected all requested changes to the article. It can be accepted as is.
Author Response
We thank the reviewer for their approval and for their valuable feedback throughout the review process.
Reviewer 2 Report
Comments and Suggestions for Authors
I would like to thank the authors for taking my suggestions into account and for proofreading the manuscript. The article now looks much better. I would like to suggest one more minor change concerning Table 2. Currently, the order is Clinican, DermFlow, Claude. In my opinion, better readability would be achieved by ordering from general to specific, taking into account the role of Clinican as a reference point. I suggest adopting the following order of columns: Claude, DermFlow, Clinican. This will allow for comparison of similar aspects of classification quality, but will ensure a more easily understandable order.
Author Response
Comment 1: "I would like to thank the authors for taking my suggestions into account and for proofreading the manuscript. The article now looks much better. I would like to suggest one more minor change concerning Table 2. Currently, the order is Clinican, DermFlow, Claude. In my opinion, better readability would be achieved by ordering from general to specific, taking into account the role of Clinican as a reference point. I suggest adopting the following order of columns: Claude, DermFlow, Clinican. This will allow for comparison of similar aspects of classification quality, but will ensure a more easily understandable order."
Response 1: We sincerely thank the reviewer for their continued thoughtful engagement with our manuscript and for this excellent suggestion regarding Table 2 column ordering. We agree that reordering the columns to Claude, DermFlow, Clinician improves readability and logical flow.
We have implemented this change in Table 2 and have also applied the same column ordering to Tables 4 and 5 and Figures 2, 3, and 4 for consistency throughout the manuscript. We appreciate the reviewer's attention to detail in improving the manuscript's clarity and presentation.
Reviewer 3 Report
Comments and Suggestions for Authors
Thank you for taking into account my suggestions/considerations! Congratulations on the excellent work! The manuscript is suitable for publication!
Author Response

(The authors gave the same response as above.)
